# Undiagnosed hypertension and associated factors among bank workers in Bahir Dar City, Northwest, Ethiopia, 2020. A cross-sectional study

Mekdes Dejenie[1], Sitotaw Kerie[2]*, Kidist Reba[2]

1 Department of Nursing, Debre Tabor Health Science College, Debre Tabor, Ethiopia, 2 Department of Nursing, College of Medicine and Health Sciences, Bahir Dar University, Bahir Dar, Ethiopia

ʘ These authors contributed equally to this work.

* sitkere5@gmail.com

**Data Availability Statement:** All relevant data are within the paper and its Supporting Information files.

## Abstract

### Background

Undiagnosed hypertension is defined as individuals who were hypertensive but did not report having been told by a health professional that they have hypertension. It is an important risk factor for development of chronic kidney disease, cardiovascular disease and all-cause mortality. Despite those problems and benefits of finding individuals with undiagnosed hypertension to prevent its outcomes, no enough investigations have been done regarding the prevalence and associated factors of undiagnosed hypertension. Therefore, the objective of this study was to assess the prevalence of undiagnosed hypertension and its associated factors among bank workers in Ethiopia.

### Methods

An institutional based cross-sectional study was held at Bahir Dar city bank workers. The simple random sampling technique was used to select the study participants. Self-administered structured questionnaire and physical measurement were used to collect data. The data were entered into EPI data 3.1 versions and exported to SPSS version 23.0 statistical software for further analysis. In order to decide the association between independent and dependent variables; multivariate logistic regression analysis was implemented. A P-value of < 0.05 was used as the criterion for statistical significance and OR with 95% confidence interval was used to indicate the strength of association.

### Result

In this study from a total of 524 participants 513 were completed the questionnaire correctly, which gives a response rate of 97.9%. The overall prevalence of undiagnosed hypertension among bank workers was 24.8% with (95% CI 21.1–28.5). Multivariate logistic regression revealed that age 35–44 [AOR = 2.56, 95% CI: (1.60–4.09)], being male [AOR = 3.61, 95% CI: (1.84–7.05)], having moderate knowledge [AOR = 3.81, 95% CI: (2.29–6.34)], having

**Funding:** This study was funded by Amhara regional health bureau.

**Competing interests:** The authors have declared that no competing interests exist.

**Abbreviations:** AOR, Adjusted Odds Ratio; BMI, Body Mass Index; CI, Confidence Interval; NCD, Non-Communicable Disease; P/E, Physical Exercise; SD, Standard Deviation; WHO, World Health Organization; Wt, Weight.

poor knowledge [AOR = 6.19, 95% CI: (3.07-)12.48], and being physically inactivity [AOR = 2.91, 95% CI: (1.26–6.76)] were variables significantly associated with undiagnosed hypertension.

## Conclusion

The prevalence of undiagnosed hypertension among bank workers in Bahir Dar city was found to be high. An age group of 35–44 years, being male, having moderate and poor knowledge and being physically inactivity was the variables that were significantly associated with undiagnosed hypertension. Therefore, creating awareness, frequent screening and implementation of an appropriate intervention for this vulnerable group is important.

## Introduction

Hypertension is defined as two or more readings of systolic blood pressure measurement of 130 mm Hg or higher or diastolic blood pressure measurement of 80 mm Hg or higher [1]. Globally, around one billion people are affected by hypertension and it is predicted to increase to 1.5 billion by 2025 [2].

Hypertension accounts for an estimated 54 percent of all strokes, 47 percent of all ischemic heart disease and 7·6 million premature deaths universally [3]. It rarely causes symptoms in the early stages and many people go undiagnosed [4].

Undiagnosed hypertension is defined as individuals who were hypertensive but did not report having been told by a health professional that they have hypertension. It is an important risk factor for development of chronic kidney disease, cardiovascular disease and all-cause mortality [5].

In 2007, around 50% of the world population were living with undiagnosed hypertension [6]. One out of three adults have hypertension and more than 50% of them are unaware of their status [7].The burden of undiagnosed hypertension increases with age increases, it ranges from 6.0% in the age group of 18–19 years to 28.7% in the age group of 65–69 years [8]. Its prevalence is also significantly higher in the rural areas 20.7% compared to urban areas 16.1% and regarding sex it is higher in males 18.6% than in females 15.6% [9].

The number of people with hypertension who are undiagnosed, untreated and uncontrolled are higher in low- and middle-income countries than developed countries [4]. In Sub Saharan Africa a large proportion of the population with hypertension remains undiagnosed, untreated, or inadequately treated, and this results in a significant burden on health, economic, and cardiovascular disease [10]. Poor access to health information and services and low socio-economic status were some of the factors contribute to the high prevalence of undiagnosed hypertension in the region [10].

In Ethiopia, the magnitude of undiagnosed hypertension is 15.6% and only a very small percentage of people had been aware of their high blood pressure [11].

Different studies conducted in different areas revealed that younger and older age, lower socioeconomic status, drinking alcohol, being underweight, absence of associated cardiovascular co-morbidities, no familial history of hypertension and primary educated individuals are more likely to have undiagnosed hypertension [12–17].

Even though, some studies have been conducted on the assessment of the prevalence and associated factors of undiagnosed hypertension in different countries. In Ethiopia, there are only few studies conducted regarding the prevalence of undiagnosed hypertension and its

associated factors in the general community. However, bank workers are at greater risk for cardiovascular diseases than the general community because of using mechanized transportations and prolonged sitting due to their job behaviors [18], there is no research conducted among bank workers to identify the prevalence and associated factors of undiagnosed hypertension in Ethiopia [19]. Therefore, this study was conducted to assess the prevalence and associated factors of undiagnosed hypertension among bank workers in Bahir Dar city to fill the gap.

## Methods and materials

### Study design and period

An institutional based cross-sectional study was conducted on 513 participants in Bahir Dar city Northwest Ethiopia from February 24 to March 24, 2020. The site is located 565 km northwest of Addis Ababa and it is the capital city of Amhara regional state. There are 2 governmental and 16 private banks with 624 and 995 workers respectively in the city.

### Sample size calculation

The number of samples required for this study was calculated for each specific objective by considering double population proportion formula by using Epi-Info version-7.2 for associated factors and the single population proportion formula was employed for the dependent variable. For single population proportion formula, the following assumptions were considered.

$$n = \frac{[((z\,a/2)]^2\,p\,(1-p))}{d^2}$$

Where: n = the required sample size

$\alpha$ = level of significance (0.05)

Z = the standard normal distribution with 95% CI (1.96)

P = prevalence of undiagnosed hypertension (13.25%)

d = tolerable margin of error (d) = 0.05

➢ n = $(1.96)^2$ 0.1325(1–0.1325)/ $(0.05)^2$ = 177

▪ Adding 10% non-response rate and with design effect of 2 = 390

For the second objective sample size was determined by using double population proportion formula and two key factors were taken from the previous literature and sample size was computed by Epi info version 7.2 software (**Table 1**).

Thus the required sample size of this study was determined by taking the maximum sample size from the second objective. Therefore the final sample size for this study was 524 bank workers.

**Table 1. Sample size calculation by using different variables.**

| S. no | Associated factors | Assumptions | The final sample size |
|---|---|---|---|
| 1 | BMI (body mass index) | Power = 80%, Ratio = 1:1, Outcome in unexposed group = 12.52%, AOR = 2.7, Outcome in exposed group = 27.9% and adding 10% non-response rate and with design effect of 2. | 524 |
| 2 | Alcohol drinking | Power = 80%, Ratio = 1:1, Outcome in unexposed group = 20.37%, AOR = 2.9, Outcome in exposed group = 42.6% and adding 10% non-response rate and with design effect of 2. | 338 |

## Participants

Bahir Dar city has 18 banks with a total of 1,697 workers. All bank workers in the city were the source population of this study. Bank workers who were working in the selected banks and available during the time of data collection and those who were not diagnosed as hypertensive and/or use of anti-hypertensive medications were included from the study. Pregnant bank workers and Janitors were excluded from the study. Nine banks with a total of 916 workers were selected by using the lottery method. To select the study participants from each selected bank, first the list of all bank workers were obtained from each selected bank. Secondly, each member was numbered or assigned a sequential number. The sample size was proportionally allocated for each bank. Finally, a total of 524 bank workers were selected by using the random generator software.

## Data collection and quality control

The data collection has two components: first there were questionnaires to collect socio-demographic, behavioral characteristics, history of chronic illnesses, hypertension related knowledge questions. Second, measurements of weight, height and blood pressure.

The questionnaire is adapted from previous similar studies and the WHO STEPS wise approach guidelines on NCD risk factor surveillance questionnaire. It contains information about socio demographic characteristics, behavioral characteristics, history of chronic illnesses, hypertension related knowledge questions and weight, height and blood pressure measurements [6, 13, 20–30]

Auscultatory method of BP measurement was used. Two measurements were taken with a minimum of 15 minutes apart using left arm consistently and the average of two BP measurements was used to determine the status of the participant.

Firstly, they were requested to avoid caffeine for 30 minutes prior to measurement. Participants were seated quietly for 5 minutes in a chair with feet on the floor and right arm was bared and supported at heart level. A standard sphygmomanometer and a standard stethoscope were used to ensure accuracy. For manual determinations, palpated radial pulse obliteration pressure was used to estimate systolic blood pressure (SBP), the cuff was inflated 20–30 mmHg above this level for the auscultatory determinations; the cuff deflation rate for auscultatory readings was 2 mmHg per second. SBP was recorded at the point at which the first of two Korotkoff sounds was heard and the disappearance of Korotkoff sound was used to define diastolic blood pressure (DBP).

Weight was measured, in kilograms using a portable weighing scale with the subjects standing, arms hanging naturally at the sides, without footwear material that may increase the body weight of the participant. Height was measured, in meters, using a stadiometer, to the crown of the head, the subject standing without any footwear or headgear and looking straight ahead.

Then body mass index was calculated by using the formula (weight in Kg/ height in m2) and classified based on the WHO classification [31]. Four BSc trained nurses were participated to collect the data.

Two days training was given for data collectors about data collection techniques and measurement procedures. Pre-test was done on (5%) of our sample size at Debre Tabor town bank workers who were not part of the study participant. The data collectors were supervised daily and the collected data were checked daily by the principal investigators for completeness. Blood pressure and physical measures were done by using standard measurement tools.

## Measurement

Hypertension was measured based on the American Heart Association Hypertension Guideline [1]. It was developed by the American college of cardiology and American heart

association, in 2017 for the prevention, detection, evaluation, and management of high blood pressure in adults.

According to this guideline, the level of blood pressure is classified as Normal; systolic blood pressure <120 and diastolic blood pressure <80 mm Hg, Elevated; systolic blood pressure 120–129 mm Hg and diastolic blood pressure <80 mm Hg, Hypertension: stage 1; systolic blood pressure 130–139 mm Hg or diastolic blood pressure 80–89 mm Hg and Hypertension: stage 2; systolic blood pressure ≥140 mm Hg or diastolic blood pressure ≥90 mm Hg. Knowledge about hypertension was assessed by questionnaires adapted from the previous studies. Level of knowledge was categorized as Good: Knowledge scores 80% and above, Moderate: Knowledge scores between 60 and 79% and Poor: Knowledge scores below 60% [6, 13, 20–28, 32]. Personal behavior and clinical related variables were assessed by using WHO STEPS wise approach guidelines on chronic risk factor surveillance questionnaire, which was developed in 2005 [30].

## Statistical analysis

The data were cleaned, coded and entered to Epi Data version 3.1 for further organizing and processing. Then, transported to IBM SPSS version 23 for analysis. The first association between each independent variable and dependent variable was assessed in bivariable analyses. Then, those independent variables with P value < 0.25 were transported to multivariate logistic regression to control the cofounders and to identify predictors of hypertension. A P-value of < 0.05 was used as the criterion for statistical significance and OR with 95%confidence interval was used to indicate the strength of association. Model fitness was tested by the Hosmer and lemeshow goodness of fit test (P-value = 0.130).

## Ethical consideration

Ethical clearance was obtained from an ethical review committee of Bahir Dar University, College of Medicine and Health Sciences. The letter was obtained from the department of adult health nursing to each selected bank. During the data collection time the aim of the study was explained and written informed consent was obtained from study participants. Confidentiality of the information was assured throughout the data collection process. There was no invasive procedure performed to conduct this study, instead measuring of physical composition was performed. This really consumed their time and to some extent disturbed their participants. Participants with increased blood pressure were advised regarding appropriate medical care.

## Results

### Socio demographic characteristics of the respondents

Of the total 524 eligible respondents, 513 respondents were complete the study with a response rate of 97.9%. Of the respondents 394 (76.8%) were males, 284 (55.4%) were age ranged from 20 to 34 years, with a mean age of 34.1 years (SD±6. 6), regarding to the educational status 293 (57.1%) were educated at degree. About 309 (60.2%) were married, 484 (94.3%) were Orthodox Christianity followers and 370 (72.1%) had ≤ 10-year work experience (**Table 2**).

### Behavioral characteristics of the participants

Among respondents, none of them were smoking cigarettes, 11 (2.1%) of them were chewing khat. Three hundred thirty-seven (65.7%) of respondents were drinking alcohol, of those, two hundred ninety-eight (58.1%) of respondents were eating fruits, 407 (79.3%) of respondents

**Table 2. Socio-demographic characteristics of bank workers in Bahir Dar city, Northwest, Ethiopia, 2020 (n = 513).**

| Variables | | Frequency | Percentage (%) |
|---|---|---|---|
| Age | 20–34 | 284 | 55.4 |
| | 35–44 | 217 | 42.3 |
| | ≥45 | 12 | 2.3 |
| Sex | Male | 394 | 76.8 |
| | Female | 119 | 23.2 |
| Marital status | Single | 194 | 37.8 |
| | Married | 309 | 60.2 |
| | Divorced | 10 | 2.0 |
| Educational level | High school | 36 | 7 |
| | Diploma | 40 | 7.8 |
| | Graduate | 293 | 57.1 |
| | Post graduate | 144 | 28.1 |
| Job description | Manager | 38 | 7.4 |
| | Officer | 380 | 74.1 |
| | Clerical | 21 | 4.1 |
| | Guard | 74 | 14.4 |
| Working experience | ≤ 10 years | 370 | 72.1 |
| | >10 years | 143 | 27.9 |

were eating vegetables and 119 (23.2%) of respondents were performed physical exercise (Table 3).

**Table 3. Behavioral characteristics of bank workers in Bahir Dar city, Northwest, Ethiopia, 2020 (n = 513).**

| Variables | | Frequency (%) | | Number of drinks | Sex | |
|---|---|---|---|---|---|---|
| | | | | | Male | Female |
| **Alcohol drinking** | Yes | 337 (65.7%) | | Less than 1 drink | 30 (8.9%) | 22 (6.5%) |
| | | | | 1–3 drinks | 159 (47.2%) | 10 (3.0%) |
| | | | | 4–6 drinks | 116 (34.4%) | (0.0%) |
| | No | 176 (34.3%) | | | | |
| **Eating fruits** | Yes | 298 (58.1%) | Frequency | Daily | 57 (19.1%) | |
| | | | | 1–4 d/wk | 241 (80.9%) | |
| | | | Serving | 1–4 servings | 262 (87.9%) | |
| | | | | ≥5 servings | 36 (20.1%) | |
| | No | 215 (41.9%) | | | | |
| **Eating vegetables** | Yes | 407 (79.3%) | Frequency | Daily | 45 (11%) | |
| | | | | 1–4 d/wk | 362 (89%) | |
| | | | Serving | 1–4 servings | 372 (91.4%) | |
| | | | | ≥5 servings | 35 (8.6%) | |
| | No | 106 (20.7%) | | | | |
| **Performing P/E** | Yes | 119 (23.2%) | frequency | <5 d/wk | 43 (36.1%) | |
| | | | | ≥5 d/wk | 76 (63.9%) | |
| | | | duration | <30 m | 47 (39.5%) | |
| | | | | ≥30 m | 72 (60.5%) | |
| | No | 394 (76.8%) | | | | |

P/E = physical exercise, d/wk = day per week, m = minute

### Clinical related characteristics

All study participants do not have a history of cardiovascular disease, sixty-five (12.7%) of respondents have a family history of hypertension, eleven (2.1%) of respondents have a history of diabetes mellitus and seven (1.4%) of respondents have a history of kidney problem. Regarding body mass index the majority 381 (74.3%) of respondents have a normal body mass index, 4.7% of them were underweight, 18.5% were overweight and 2.5% were obese.

### Knowledge of participants regarding hypertension

The majority 235 (45.8%) of respondents had a good level of knowledge, 42.9% had moderate level of knowledge and 45.8% had poor level of knowledge regarding hypertension.

### Blood pressure status of respondents

Out of 513 participants 174 (33.9%) have normal blood pressure, 212 (41.3%) have elevated blood pressure, and 127 (24.8%) was hypertensive. Among those hypertensive individuals, 69 (13.5%) have stage 1 and 58 (11.3%) have stage 2 hypertension.

### Factors associated with undiagnosed hypertension

All independent variables were analyzed in the bivariable analysis. Of all variables age, sex, marital status, educational level, job description, knowledge about hypertension, consuming vegetables, performing regular physical exercise and body mass index were included in the multivariable analysis. In the multivariable logistic regression analysis age, sex, hypertension knowledge and performing regular physical exercise were significantly associated with undiagnosed hypertension (P-values < 0.05).

Respondents with age group of 35–44 were 2.56 times more likely to have undiagnosed hypertension as compared with age group of 20–34 (AOR = 2.56, 95% CI: 1.60–4.18), males were 3.61 times more likely to have undiagnosed hypertension as compared with females (AOR = 3.61, 95% CI: 1.84–7.05). Those who had moderate knowledge about hypertension were 3.81 times more likely to have undiagnosed hypertension as compared with those who had good knowledge with (AOR = 3.81, 95% CI: 2.29–6.34). And those who had poor knowledge about hypertension were 6.19 times more likely to have undiagnosed hypertension as compared with those who had good knowledge with (AOR = 6.19, 95% CI: 3.19–12.48). Those who had not performed regular physical exercise were 2.91 times more likely to have undiagnosed hypertension than those who were performing regular physical exercise with (AOR 2.91, 95% CI: 1.26–6.76) **(Table 4).**

## Discussion

This an institutional based cross-sectional study with the objective of the assessment of the prevalence of undiagnosed hypertension and its associated factors among bank workers in Bahir Dar city, Northwest Ethiopia was assessed the magnitude of undiagnosed hypertension and associated factors.

This study revealed that the magnitude of undiagnosed hypertension among bank workers was found to be 24.8% with (95% CI 21–29). This finding is consistent with studies done in Western India, 26% [33], Nigeria 25% [34], rural area of West Bengal 24.1% [35], and in Finland 24% [36].

On the other hand, the result of this study is higher than studies conducted in the Byblos, Lebanon 16.9% [13], in India 10.1% [37], in United States of America 19.7% [38], in Iran 4.8% [21], in Ghana 18.5% [39], in Hosanna 10.2% [40], in Addis Ababa 13.25% [41] and Hawassa 12.3% [42].

**Table 4. The bivariable and multivariable logistic regression analysis for factors associated with undiagnosed hypertension among bank workers in Bahir Dar city, 2020.**

| Variables | | Undiagnosed Hypertension | | | |
|---|---|---|---|---|---|
| | | Yes | No | COR (95% CI) | AOR (95% CI) |
| Age | ≥45 | 5 (0.6) | 7 (1.4) | 3.52 (1.07–11.53) * | 2.32(0.63–8.63) |
| | 35–44 | 74 (14.6) | 143 (27.9) | 2.54 (1.67–3.87) ** | 2.56 (1.60–4.18) ** |
| | 20–34 | 48 (9.6) | 236 (46) | 1 | 1 |
| Sex | Male | 115 (22.4) | 279 (54.4) | 3.68(1.95–6.94) ** | 3.61(1.84–7.05) ** |
| | Female | 12 (2.3) | 107 (20.9) | 1 | 1 |
| Marital status | Single | 39 (7.6) | 155 (30.2) | 0.25 (0.07–0.91) | 0.47 (0.10–2.14) |
| | Married | 83 (16.4) | 226 (44.1) | 0.37(0.10–1.30) | 0.54 (0.13–2.34) |
| | Divorced | 5 (0.8) | 5 (1.0) | 1 | 1 |
| Educational level | H/school | 6 (1.2) | 30 (5.8) | 0.50(0.25–1.37) | 0.35 (0.15–1.40) |
| | Diploma | 6 (1.2) | 34 (6.6) | 0.44 (0.17–1.14) | 0.54 (0.15–1.96) |
| | Degree | 74 (14.4) | 219 (42.7) | 0.85 (0.54–1.33) | 1.01(0.58–1.75) |
| Job description | Manager | 16 (3.1) | 22 (4.3) | 2.86 (1.21–6.75) * | 2.03 (0.78–5.30) |
| | Officer | 90 (18.5) | 290 (55.4) | 1.22(0.66–2.26) | 1.20 (0.68–2.37) |
| | Clerical | 6 (0.2) | 15 (4.1) | 1.57 (0.52–4.74) | 2.32 (0.62–8.44) |
| | Guard | 15 (2.9) | 59 (11.) | 1 | 1 |
| Knowledge level | Poor | 25 (4.9) | 33 (6.4) | 4.99 (2.62–9.48) ** | 6.20(3.07–12.59) ** |
| | Moderate | 71 (13.8) | 149 (29) | 3.14 (1.96–5.03) ** | 3.81 (2.35–6.34) ** |
| | Good | 31 (6.0) | 204 (39.8) | 1 | 1 |
| Consuming vegetables | Not consume | 33 (6.4) | 73 (14.2) | 2.71(0.97–7.61) | 2.00(0.65–6.22) |
| | Low | 89 (17.5) | 283 (55.2) | 1.89 (.71–5.01) | 1.56 (0.54–4.54) |
| | Normal | 5 (0.8) | 30 (5.8) | 1 | 1 |
| Performing regular P/E | Yes | 7 (1.4) | 65 (12.7) | 1 | 1 |
| | No | 120 (23.4) | 321 (62.6) | 3.47(1.55–7.78) * | 2.91 (1.26–6.76) * |
| BMI | Under Wt | 5 (0.6) | 19 (4.1) | 1 | 1 |
| | Normal Wt | 94 (18.7) | 287 (55.6) | 1.25 (0.45–3.43) | 0.71 (0.24–2.15) |
| | Over Wt | 22 (5.1) | 73 (13.5) | 1.15(0.38–3.42) | 0.78 (0.24–2.69) |
| | Obese | 6 (0.4) | 7 (2.1) | 3.26(0.75–14.16) | 1.27 (0.25–6.50) |

* = P-value < 0.05,

** = P-value <0.01, P/E = Physical exercise, BMI = body mass index, Wt = weight

This discrepancy may be due to the difference study participants. In this study only bank workers were involved as a study participant and the nature of work can dispose of them to a sedentary life and varying level of stress. Sedentary life cause hypertension by increasing atherosclerosis and stress also causes high blood pressure by causing the secretions of hormones that has vasoconstriction effect [43]. The other possible reason for the discrepancy may be due to the use of different guidelines to define hypertension. This study uses the new hypertension guideline to define hypertension, as any systolic BP measurement of 130 mm Hg or higher or any diastolic BP measurement of 80 mm Hg or higher [1].While, the previous studies used the previous guideline to define hypertension as any systolic BP measurement of 140 mm Hg or higher or any diastolic BP measurement of 90 mm Hg or higher. The new definition of hypertension contributes to high prevalence of undiagnosed hypertension in the current study.

But this finding was lower than from studies done in Bangladeshi 59.9% [17], in Nigeria 36.1% [44], in Central Province of Sri Lanka 31.7% [45], in Ireland 41.2% [24], and in Sudan

38.2% [46]. This discrepancy may be due to the study subject and sociodemographic differences; the current study was conducted among the banking workers with the mean age of 34. 1 ±6. 6 and includes both sexes. Whereas, the study conducted in Bangladeshi was among patients with age greater than 35 years [17], and studies in Nigeria and Ireland were conducted among the participants with age 40±8. 5 and 50+ years, respectively [44, 45]. And also, a study done in Central Province of Sri Lanka was included only males as a participant [47]. Being sick, older age and male can contribute to the prevalence of high blood pressure.

In this study age, sex, knowledge about hypertension and regular physical exercise were found to be significantly associated with undiagnosed hypertension.

Respondents with the age group of 35–44 were 2.56 times more likely to have undiagnosed hypertension as compared with age group of 20–34. This finding was supported by a study done in Malaysia, in north central Nigeria, in Sudan, in Hosanna and in Addis Ababa [12, 21, 40, 41, 48]. It is known that arterial stiffness become increased with age, which will contribute to the higher prevalence of hypertension [49].

Sex was also another variable which was significantly associated with undiagnosed hypertension. Male participants were 3.61 times more likely to have undiagnosed hypertension than females. This finding was supported by the previous studies conducted in Bangladeshi and North West of Iran [17, 50]. And women had a higher rate of getting their blood pressure checked than men [51]. Women have more chance to get frequent health services like family planning, antenatal care, delivery and immunization and have to visit health professionals. This creates an opportunity to early diagnose with some typical health screening, including hypertension. On the other hand, males are more alcohol drinker than women and this contributes for hypertension and negligence for early diagnosed.

Respondents with poor and moderate knowledge about hypertension were 6.2 and 3.81 times more likely to have undiagnosed hypertension as compared with those who had good knowledge respectively. This finding was supported by the study conducted in rural Rwanda and Cracow [6, 52]. The respondents with good knowledge about hypertension might have a better healthy lifestyle and health-seeking behavior.

Physical exercise was also another variable which had a significant association with undiagnosed hypertension. Respondents who did not perform regular physical exercise were 2.91 times more likely to have undiagnosed hypertension than those respondents who perform regular physical exercise. This finding was supported by the previous study conducted in Hawassa [42]. Physical exercise can prevent blood pressure by reducing blood vessel stiffness [53].

The possible limitations of this study are; first, there was not pregnancy test for female participants and females with early pregnancy had chance to include in this study. Second, community-based studies were used for sample size calculation and for comparison of the results of this institutional-based study. Third, substance use related factors were not assessed by standard tool. Fourth, using of old version WHO STEPS guideline to assess personal behavior and clinical related variables.

## Conclusion

The prevalence of undiagnosed hypertension among bank workers in Bahir Dar city northwest Ethiopia was found to be high. Being age group of 35–44 years, being male, having poor and moderate knowledge about hypertension and being physically inactivity was significantly associated with undiagnosed hypertension. Therefore, frequent screening and creating awareness on asymptomatic nature of hypertension and the benefit of physical exercise for preventing high blood pressure for bank workers is significant.

## Supporting information

**S1 Data.**
(SAV)

**S1 Questionnaire.**
(DOCX)

## Acknowledgments

The authors are grateful to the participants who participated in the study.

## Author Contributions

**Conceptualization:** Mekdes Dejenie, Kidist Reba.

**Data curation:** Mekdes Dejenie.

**Formal analysis:** Sitotaw Kerie.

**Methodology:** Sitotaw Kerie.

**Software:** Sitotaw Kerie.

**Supervision:** Kidist Reba.

**Validation:** Sitotaw Kerie.

**Writing – original draft:** Mekdes Dejenie, Sitotaw Kerie.

**Writing – review & editing:** Kidist Reba.

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
