## [Decision Letter · Decision Letter 0]

14 Dec 2020

PONE-D-20-28178

Undiagnosed Hypertension and Associated Factors Among Bank Workers in Bahir Dar City, Northwest, Ethiopia, 2020. A Cross-Sectional Study.

PLOS ONE

Dear Dr. Kerie,

Thank you for submitting your manuscript to PLOS ONE. After careful consideration, we feel that it has merit but does not fully meet PLOS ONE’s publication criteria as it currently stands. Therefore, we invite you to submit a revised version of the manuscript that addresses the points raised during the review process.

Specific feedback is provided below to improve the quality of the manuscript. 

We look forward to receiving your revised manuscript.

Kind regards,

Yvonne Commodore-Mensah

Academic Editor

PLOS ONE

Journal Requirements:

3. Please include additional information regarding the survey or questionnaire used in the study and ensure that you have provided sufficient details that others could replicate the analyses. For instance, if you developed a questionnaire as part of this study and it is not under a copyright more restrictive than CC-BY, please include a copy, in both the original language and English, as Supporting Information.  If the original language is written in non-Latin characters, for example Amharic, Chinese, or Korean, please use a file format that ensures these characters are visible.

4. Please state whether you validated the questionnaire prior to testing on study participants. Please provide details regarding the validation group within the methods section.

5. In your Methods section, please provide additional information about the participant recruitment method and the demographic details of your participants. Please ensure you have provided sufficient details to replicate the analyses such as:   

-    a description of any inclusion/exclusion criteria that were applied to participant recruitment,

-    a statement as to whether your sample can be considered representative of a larger population,

-    a description of how participants were recruited, and

-       descriptions of where participants were recruited and where the research took place.

6. We note that you have indicated that data from this study are available upon request. PLOS only allows data to be available upon request if there are legal or ethical restrictions on sharing data publicly. For information on unacceptable data access restrictions, please see http://journals.plos.org/plosone/s/data-availability#loc-unacceptable-data-access-restrictions.

7. Thank you for stating the following beneath the Acknowledgments Section of your manuscript:

'Funding

This study was supported by Amhara Regional Health Bureau.'

'No. The funders had no role in study design, data collection and analysis, decision to publish, or preparation of the manuscript'

Additional Editor Comments:

The article would benefit from a thorough editorial review for grammatical errors in order to improve the quality of the manuscript.

Abstract

• Specify that study was conducted in Ethiopia

• Specify the statistical approach that was used for the analysis in the last sentence of the methods section

• What percentage of participants had diagnosed hypertension?

• Round up the adjusted odds ratios and confidence intervals to 2 decimal places.

Introduction

• Page 4, line 75, revise sentence from “In the year of 2007” to “In 2007”

• Line 90 and 89 are redundant. Both sentences can be combined to make the point more concise

The study participants

• Change the subheading to Participants

• Provide range or approximate number of bank workers to provide sufficient information on the source population.

Sample size and procedure

• The formula for the sample size calculation is not needed in this section. Table 1 is also not necessary.

• Reference should be provided for WHO classification

Measurement

• The first sentence should be revised and the correct description should be used to describe the American Heart Association Hypertension Clinical Guidelines. A citation should be provided after the first sentence.

• The last sentence is unclear. What guideline is being referred to in this context?

Data quality control

• This section can be integrated into the other sections and shortened.

• There is a typo in line 183 “questioner”

Statistical Analysis

• Add description

Results

• Line 222: add a description of chat for readers who are not familiar with that terminology

• There are several grammatical errors that need to be corrected.

• In Table 3, the number of drinks should be presented separately for men and women because of the differences in recommendation.

• Add footnote should be added to Table 3 to provide a description of all the abbreviations used in Table 3.

• All Odds Ratios and 95%CIs should be reported to two decimal places.

• The proportion of participants who had normal blood pressure, elevated blood pressure, stage 1 and stage 2 hypertension are not reported.

Reviewers' comments:

Reviewer's Responses to Questions

**Comments to the Author**

1. Is the manuscript technically sound, and do the data support the conclusions?

Reviewer #1: Yes

2. Has the statistical analysis been performed appropriately and rigorously? 

Reviewer #1: Yes

3. Have the authors made all data underlying the findings in their manuscript fully available?

Reviewer #1: Yes

4. Is the manuscript presented in an intelligible fashion and written in standard English?

Reviewer #1: No

5. Review Comments to the Author

Reviewer #1: 1. Introduction

On the introduction, the reason for choosing bank workers from the general population for the study should be emphasized.

2. Materials and Methods

Sample size and procedure

On the study design and period section, the authors have mentioned that there are 2 government and 16 private banks in Bahir Dar, how many branches were included in the study? What was your selection criteria?

The authors use P value from a study conducted in the general population in Gulelle sub-city of Addis Ababa while your study was focused on a specific segment of the population. How do you explain this?

Measurement

Line 169-174

“Level of blood pressure is classified as Normal; systolic blood pressure <120 and diastolic blood pressure <80 mm Hg, Elevated; systolic blood pressure 120-129 mm Hg and diastolic blood pressure <80 mm Hg, Hypertension: stage 1; systolic blood pressure 130-139 mm Hg or diastolic blood pressure 80-89 mm Hg and Hypertension: stage 2; systolic blood pressure =140 mm Hg or diastolic blood pressure =90 mm Hg”. Such classification is not used in the result section. Better to present information in a uniform manner.

The STEPS manual have an updated version as of 26 January 2017, why did you stick to the older version?

Data analysis

The P-value cutoff to declare significance mentioned in the methodology section is inconsistent with that in the result section.

Results

Table 2: Marital status…….check for percentage.

Table 3: the percentage for drinking amount, frequency of eating fruit, serving of fruits, frequency of eating vegetables, serving of vegetables, exercise frequency, exercise duration should be calculated out of 100%.

Factors associated with undiagnosed hypertension: in the logistic regression analysis, how did you code for “yes” and “no” of undiagnosed hypertension? as there is a miss match between the COR and the frequency placed under each variable.

Discussion

Line 301: can we categorize the age group of 35-44 in the old age group?

6. PLOS authors have the option to publish the peer review history of their article (what does this mean?). If published, this will include your full peer review and any attached files.

Reviewer #1: **Yes: **Firehiwot Amare

---

## [Author Response · Author response to Decision Letter 0]

28 Dec 2020

I have uploaded the response to reviewers with the title "Response to Reviewers"'

---

## [Decision Letter · Decision Letter 1]

6 May 2021

PONE-D-20-28178R1

Undiagnosed Hypertension And Associated Factors Among Bank Workers In Bahir Dar City, Northwest, Ethiopia, 2020. A Cross-Sectional Study.

PLOS ONE

Dear Dr. Kerie,

Thank you for submitting your manuscript to PLOS ONE.  After review, there are a few changes to the manuscript that the referee has suggested, including a better presentation of the sample size calculation. Therefore, we invite you to submit a revised version of the manuscript that addresses the points raised during the review process.

We look forward to receiving your revised manuscript.

Kind regards,

Colin Johnson, Ph.D.

Academic Editor

PLOS ONE

Journal Requirements:

Reviewers' comments:

Reviewer's Responses to Questions

**Comments to the Author**

1. If the authors have adequately addressed your comments raised in a previous round of review and you feel that this manuscript is now acceptable for publication, you may indicate that here to bypass the “Comments to the Author” section, enter your conflict of interest statement in the “Confidential to Editor” section, and submit your "Accept" recommendation.

Reviewer #1: All comments have been addressed

2. Is the manuscript technically sound, and do the data support the conclusions?

Reviewer #1: Yes

3. Has the statistical analysis been performed appropriately and rigorously? 

Reviewer #1: Yes

4. Have the authors made all data underlying the findings in their manuscript fully available?

Reviewer #1: Yes

5. Is the manuscript presented in an intelligible fashion and written in standard English?

Reviewer #1: No

6. Review Comments to the Author

Reviewer #1: Abstract

Methods: line 37 “multiple logistic regression analysis was implemented” multiple should be replaced with multivariate.

Methods and materials

Participants:

Line 121 better to replace study subjects with study participants.

Line 124 it’s not clear whether the proportional allocation preceded or succeeded the selection of the bank workers by the random generator software.

Sample size calculation should not be deleted instead presented in a precise manner (I don’t think the recommendation of the previous reviewer was for you to delete the whole thing)

7. PLOS authors have the option to publish the peer review history of their article (what does this mean?). If published, this will include your full peer review and any attached files.

Reviewer #1: **Yes: **Firehiwot Amare (B.Pharm; M.Pharm)

---

## [Author Response · Author response to Decision Letter 1]

10 May 2021

We have uploaded the responses with the title of cover letter

---

## [Editor Report · Decision Letter 2]

14 May 2021

Undiagnosed Hypertension And Associated Factors Among Bank Workers In Bahir Dar City, Northwest, Ethiopia, 2020. A Cross-Sectional Study.

PONE-D-20-28178R2

Dear Dr. Kerie,

We’re pleased to inform you that your manuscript has been judged scientifically suitable for publication and will be formally accepted for publication once it meets all outstanding technical requirements.

Kind regards,

Colin Johnson, Ph.D.

Academic Editor

PLOS ONE
---

## [Editor Report · Acceptance letter]

18 May 2021

PONE-D-20-28178R2 

Undiagnosed Hypertension And Associated Factors Among Bank Workers In Bahir Dar City, Northwest, Ethiopia, 2020. A Cross-Sectional Study. 

Dear Dr. Kerie:

I'm pleased to inform you that your manuscript has been deemed suitable for publication in PLOS ONE. Congratulations! Your manuscript is now with our production department. 

Kind regards, 

on behalf of

Dr. Colin Johnson 

Academic Editor

PLOS ONE